# Stochastic colonization of hosts with a finite lifespan can drive individual host microbes out of equilibrium

**Román Zapién-Campos** **, Michael Sieber** , **Arne Traulsen** *

Max Planck Institute for Evolutionary Biology, August-Thienemann-Str. 2, 24306 Plön, Germany

* traulsen@evolbio.mpg.de

## Abstract

Macroorganisms are inhabited by microbial communities that often change through the lifespan of an individual. One of the factors contributing to this change is colonization from the environment. The colonization of initially microbe-free hosts is particularly interesting, as their microbiome depends entirely on microbes of external origin. We present a mathematical model of this process with a particular emphasis on the effect of ecological drift and a finite host lifespan. Our results indicate the host lifespan becomes especially relevant for short-living organisms (e.g. *Caenorhabditis elegans*, *Drosophila melanogaster*, and *Danio rerio*). In this case, alternative microbiome states (often called enterotypes), the coexistence of microbe-free and colonized hosts, and a reduced probability of colonization can be observed in our model. These results unify multiple reported observations around colonization and suggest that no selective or deterministic drivers are necessary to explain them.

## Author summary

Microbial communities are prevalent not only in the environment but also in hosts. Although the drivers of environmental microbiomes have been studied extensively, less is known about the drivers distinguishing a host environment. Recent experimental observations have highlighted the influence of ecological drift in hosts with short lifespan, including model organisms like *C. elegans*, *D. melanogaster* and *D. rerio*. We have developed a theoretical model to study the effect of a finite host lifespan on relevant observables of the microbiome, including the microbial load, probability of colonization of a microbial taxon, and distribution of microbiome composition in a host population. Although we focus on a case free of any selection, our results indicate the possible coexistence of hosts with alternative microbiome composition, and to a larger extent the coexistence of colonized and microbe-free hosts. A quantitative description is provided.

## Introduction

Microbial communities inhabit every available habitat on this planet, including the tissues of macroorganisms. For such host-associated communities every host animal constitutes a

**Funding:** The authors received no specific funding for this work.

**Competing interests:** The authors have declared that no competing interests exist.

distinct habitat. Migration between these individual habitats and ecological drift within them play important roles in structuring these communities [1]. This idea is formalized in the Unified Neutral Theory of Biodiversity where individuals within a community are regarded as ecologically equivalent [2]. First developed in a macro-ecological context, its application has been extended to microbial populations [3, 4] and host-associated microbiomes [5–7].

What sets host-associated microbiomes apart is that their habitat—the host animal—is itself subject to demographic processes such as reproduction and death. Previous applications of neutral models to microbiome data have generally ignored these host-level processes by assuming essentially static hosts with infinite lifespans, allowing convergence to a long-term equilibrium distribution of microbial abundances, see e.g. [7]. However, any real host species will have a finite lifespan that may not allow for the community to settle down on a potential long-term composition. Moreover, differences in the lifespan across host species could obscure comparisons of neutrality across different species. Several authors have fitted one of these 'static host' neutral models [3] to microbiome datasets across multiple host species, finding an overall high resemblance [5–7]. However, one of these studies has found much less resemblance for the gut microbiome of *C. elegans* compared to sponges and hydra and speculated that this may be explained by the shorter lifetime of *C. elegans* [7]. Others have noted that the worm microbiome might be neutrally assembled, obscured by a transient state far from the neutral long-term equilibrium [8].

Few studies have explored the effect of host life cycles on the microbiome. Zeng et al. studied the change of microbiome composition under neutrality and discrete host generations, but did not consider microbial dynamics [9]. The effect of microbial symbionts, particularly the coevolution under multilevel selection [10], and the effect of the horizontal and vertical acquisition of such microbes have been studied elsewhere [11].

An additional assumption of current neutral models is that hosts contain the same abundance of microbes throughout their lives [3]. This is not the case in the gut of important model organisms like *C. elegans* [12], *D. melanogaster* [13], and *D. rerio* [14], which are initially microbe-free and only colonized from the environment later. According to the "sterile womb" hypothesis [15], also human newborns may be initially microbe-free.

By modelling the change from a microbe-free to a fully microbe-populated state, we study the transients of colonization, and their implications as the lifespan of hosts shortens. Existing models have suggested that neutral models can explain microbial abundances within hosts. Extending these neutral models of a host's microbiome to capture microbial community dynamics during the finite lifespan of a host seems thus natural. We analyse such a model, including the colonization from a microbe-free state and the finite lifespan of hosts. We discuss the dynamical consequences and the connection to experimental observations.

## Model and methods

### A nearly-neutral model

We consider multiple hosts (habitats) connected to a pool of microbes. This pool is the subset of environmental microbes capable of colonizing the hosts. Microbial abundances within each host change by three processes: (i) the death of a microbe, giving rise to empty space (ii) a birth-immigration process, when the new empty space is replaced by a microbe, and (iii) host renewal, when a host dies with its microbiome and a new host appears that does not contain any microbe. An illustration of this host-microbiome model is shown in Fig 1.

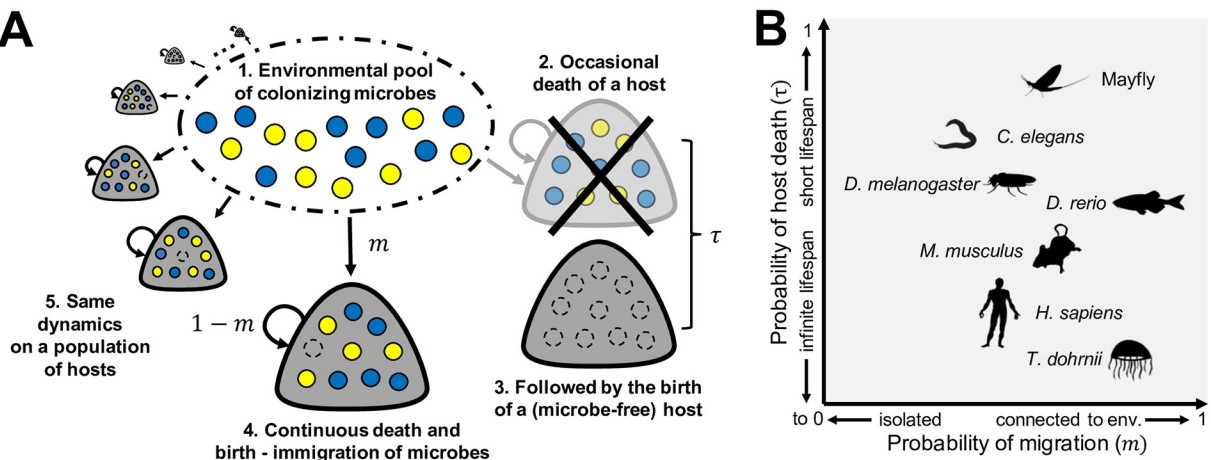

**Fig 1. The host-microbiome dynamics in our model and the coordinates of host species in a map of its parameters.** (**A**) Gray blobs indicate hosts, coloured- and empty circles indicate microbes and empty space, respectively. Within the host, microbes go through a death-birth process, with newborns migrating from a pool of colonizing microbes with probability $m$ or being chosen from the same host with probability $1-m$. The pool of microbes includes all microbes capable of living within hosts. Hosts are identical habitats, each with a finite, geometrically distributed lifespan. Each time step, there is a probability $\tau$ of a host death followed by the birth of a new host. Newborn hosts contain no microbes. The probability of a host death-birth event is relative to a microbe death-birth event. (**B**) A space of $m$ and $\tau$ provides a life-history map on top of which hosts can be located and variables interrogated (see Figs 3–6). We sketch hypothetical coordinates of multiple host species. In practice however, $m$ and $\tau$ may depend on physiological, environmental and behavioural factors. Silhouettes from PhyloPic (http://phylopic.org) licensed under Public Domain Dedication 1.0 licenses.

We consider a population of hosts that is sufficiently large to draw statistical conclusions. The microbial community in each host grows dynamically, but with a fixed maximum capacity $N$. To make this more precise, let $n_i$ be the number of individuals of the *i-th* microbial taxon within a host ($i \geq 1$) and $M$ be the number of taxa. At any time we have $\sum_{i=1}^{M} n_i \leq N$. We reserve the index $i = 0$ for the unoccupied space, namely $n_0 = N - \sum_{i=1}^{M} n_i$. We define $x_i = n_i/N$ as the frequency of the *i-th* taxon within a host and assume $N \gg 1$, such that $x_i$ is continuous and $N - 1 \approx N$. Note, that $x_0$ then denotes the fraction of available space within a host. We assume the death of hosts can be approximated as an event occurring each time step with probability $\tau$, given by the probability of host death-birth events per microbial death-birth event. The limiting case $\tau = 0$ corresponds to infinitely living hosts (as in [3, 7]), while $\tau = 1$ corresponds to hosts whose lifespan is as short as the average lifespan of a microbe, leading to almost entirely empty hosts.

Let us focus on the events within a single host. In each time step, a randomly selected site is changed. This site is either unoccupied space or a microbe. Death is followed by replacement via immigration or birth of a new type. With probability $m$, its content is replaced by a random microbe from the environment, selected proportionally to its frequency in the pool of colonizers, $p_i$ (note that $p_0 = 0$). With probability $1 - m$, it is replaced by a microbe from the same host, selected proportionally to the fitness $(1 + \alpha_i)x_i$ of the reproducing microbe—or it is replaced by unoccupied space with probability proportional to $(1 + \alpha_0)x_0$. The fitness parameter $\alpha_i$ describes deviations from strict neutrality, where proliferation of microbe $i$ is promoted ($\alpha_i > 0$) or impeded ($\alpha_i < 0$). The parameter $\alpha_0$ controls how rapidly unoccupied space within a host is filled with microbes. This determines the level of resistance a host poses to be occupied by microbes, or in other words, how favourable the host environment is for microbial reproduction and persistence. For $\alpha_0 > 0$, hosts pose an increased resistance to the internal microbes, while $\alpha_0 < 0$ decreases such resistance. The acceptable range of $\alpha_i$ and $\alpha_0$ ranges

from −1 to infinity. The resulting stochastic process for a given host can be described by the probabilities of events after one time step,

$$P[x_i \rightarrow \delta_{i,0}] \quad = \tau \tag{1a}$$

$$P\left[x_i \rightarrow x_i + \tfrac{1}{N}\right] \quad = (1-\tau)(1-x_i)\left(mp_i + (1-m)\frac{(1+\alpha_i)x_i}{\sum_j(1+\alpha_j)x_j}\right) \tag{1b}$$

$$P\left[x_i \rightarrow x_i - \tfrac{1}{N}\right] \quad = (1-\tau)x_i\left(m(1-p_i) + (1-m)\left(1-\frac{(1+\alpha_i)x_i}{\sum_j(1+\alpha_j)x_j}\right)\right) \tag{1c}$$

$$P[x_i \rightarrow x_i] \quad = 1 - P[x_i \rightarrow \delta_{i,0}] - P\left[x_i \rightarrow x_i + \tfrac{1}{N}\right] - P\left[x_i \rightarrow x_i - \tfrac{1}{N}\right], \tag{1d}$$

where Eq (1a) describes the probability of a host death event: All microbial frequencies are set to zero, i.e. $x_i \rightarrow 0$ for $i \geq 1$. At the same time, a new empty host arises, corresponding to $x_0 \rightarrow 1$. This is captured by $\delta_{i,0}$, the Kronecker delta (1 for $i = 0$ and 0 otherwise). The three other probabilities require that the host survives, which occurs with probability $1-\tau$. For a microbial taxon $i$, Eq (1b) describes the probability of increase by immigration or reproduction within the host, and Eq (1c) describes the probability of decrease derived from other taxa immigration, reproduction within the host, or their inability to reproduce. For $i = 0$, Eqs (1b) and (1c) describe the probability of increasing and decreasing the unoccupied space, respectively. Finally, Eq (1d) indicates the probability of no change. Focusing on the effect of ecological drift we fix the microbial fitness $\alpha_i = 0$ (for $i \geq 1$) for the remainder of the manuscript.

Probabilities in Eq (1) change considerably through time. For example, because hosts are largely empty at birth, unoccupied space decreases rapidly as $P\left[x_0 \rightarrow x_0 - \tfrac{1}{N}\right] \gg P\left[x_0 \rightarrow x_0 + \tfrac{1}{N}\right]$, while the microbial frequencies increase because $P\left[x_i \rightarrow x_i - \tfrac{1}{N}\right] \ll P\left[x_i \rightarrow x_i + \tfrac{1}{N}\right]$.

For $\tau = 0$ the probabilities are as in Sloan et al.'s [3], which becomes a good approximation when the time scale of reproduction on the microbial level is much faster than the time scale of reproduction on the host level. We focus on the dynamics of the probability density of $x_i$, $\Phi_i[x_i, t]$. Due to the differences in $p_i$ and $\alpha_i$, $\Phi_i[x_i, t]$ can be different for all microbial taxa $i$. This can be approximated in the large $N$ limit by a Fokker-Planck equation (see S1 Appendix), with $t$ being measured in the number of microbial death-birth events. Writing down the equations for unoccupied space $x_0$ and microbes separately we have

$$\frac{\partial}{\partial t}\Phi_0[x_0, t] = \frac{\partial}{\partial x_0}\left[-a_0[x_0]\Phi_0[x_0, t] + \frac{1}{2}\frac{\partial}{\partial x_0}b_0^2[x_0]\Phi_0[x_0, t]\right] + \tau\left(\delta_{x_0,1} - \Phi_0[x_0, t]\right) \tag{2a}$$

$$\frac{\partial}{\partial t}\Phi_i[x_i, t] = \frac{\partial}{\partial x_i}\left[-a_i[x_i]\Phi_i[x_i, t] + \frac{1}{2}\frac{\partial}{\partial x_i}b_i^2[x_i]\Phi_i[x_i, t]\right] + \tau\left(\delta_{x_i,0} - \Phi_i[x_i, t]\right), \tag{2b}$$

where $a_i[x_i]$ describes the deterministic part of the change and $b_i^2[x_i]$ describes changes due to randomness [16]. The term $a_i[x_i]$ is calculated as the first moment of $\Delta x_i$, the

expectation $\langle \Delta x_i \rangle$,

$$a_0[x_0] = (1 - \tau)\left(-mx_0 - (1-m)x_0\left(1 - \frac{1 + \alpha_0}{(1+\alpha_0)x_0 + (1-x_0)}\right)\right)\frac{1}{N} \tag{3a}$$

$$a_i[x_i] = (1 - \tau)\left(m(p_i - x_i) - (1-m)x_i\left(1 - \frac{1}{(1+\alpha_0)x_0 + (1-x_0)}\right)\right)\frac{1}{N}. \tag{3b}$$

The term $b_i^2[x_i]$ is calculated as the second moment of $\Delta x_i$, the expectation $\langle (\Delta x_i)^2 \rangle$,

$$b_0^2[x_0] = (1 - \tau)\left(mx_0 + (1-m)x_0\left(1 + \frac{(1+\alpha_0)(1-2x_0)}{(1+\alpha_0)x_0 + (1-x_0)}\right)\right)\frac{1}{N^2} \tag{4a}$$

$$b_i^2[x_i] = (1 - \tau)\left(m(p_i + x_i - 2p_ix_i) + (1-m)x_i\left(1 + \frac{(1-2x_i)}{(1+\alpha_0)x_0 + (1-x_0)}\right)\right)\frac{1}{N^2}. \tag{4b}$$

For $\tau \to 0$, the last terms in Eq (2) vanish, recovering the usual Fokker-Planck equation of the neutral model without host death [3], while for $\tau > 0$ these additional terms describe the change due to host death, where a new, microbe-free host appears.

Although individual hosts constantly change their microbiome through the process of microbial death birth-immigration and host death, the collection of transient host states becomes stationary at the population level. This stationary distribution is found setting the time derivative of Eq (2) equal to zero,

$$0 = \frac{d}{dx_0}\left[-a_0[x_0]\Phi_0[x_0] + \frac{1}{2}\frac{d}{dx_0}\left[b_0^2[x_0]\Phi_0[x_0]\right]\right] + \tau\left(\delta_{x_0,1} - \Phi_0[x_0]\right) \tag{5a}$$

$$0 = \frac{d}{dx_i}\left[-a_i[x_i]\Phi_i[x_i] + \frac{1}{2}\frac{d}{dx_i}\left[b_i^2[x_i]\Phi_i[x_i]\right]\right] + \tau\left(\delta_{x_i,0} - \Phi_i[x_i]\right) \tag{5b}$$

The Fokker-Planck approximation has several benefits: It provides an intuition of the stochastic process at the population level and the effect of host death ($\tau$), a direct connection to models not considering finite host lifespans [3], and the possibility to frame the process in the broader stochastic processes literature [16].

An alternative interpretation of the stochastic process is provided by [17]

$$\Phi_i[x_i] = \int_0^\infty \Phi_i[x_i, t_r]|_{\tau=0}\Psi[t_r]dt_r,$$

where $\Phi_i[x_i]$ results from considering all the possible distributions of the time-dependent death-birth process of microbes without host dynamics, $\Phi_i[x_i, t_r]|_{\tau=0}$, influenced by the distribution of death-birth time of hosts, $\Psi[t_r]$. The distribution of these resetting events is given by

$$\Psi[t_r] = \tau e^{-\tau t_r} \tag{6}$$

This equation will help us to compare our model and individual-based simulations.

Now we aim to solve Eq (5), where a major challenge arises from the additional terms capturing the host death-birth events, which correspond to a resetting of the local microbial community. Such resetting events are often referred to as "catastrophes" in the Mathematics literature and research has focused on finding closed form solutions of the corresponding discrete problem derived from the master equation using first order transition probabilities [18–20]. In physics, this is called diffusion-drift with resetting and its Fokker-Planck

approximation and zero order transition probabilities have been used to find closed form solutions and compute quantities of interest [17, 21]. Our model considers density-dependent transition probabilities, i.e. second order effects. Although these provide a well defined system at the boundaries $x_i = \{0, 1\}$, they complicate finding a closed form solution of $\Phi_i[x_i]$ tremendously. Approximating the solutions numerically using the finite differences and finite element methods [22] is possible.

We solved this equation numerically to query the parameter space [22]. However, we found our implementation could lead to numerical errors that were large and inconsistent in some cases, especially as $\tau \to 0$. As it proved more robust numerically (S1, S2 and S3 Figs), we used the master equation (see S1 Appendix) to produce our figures instead,

$$\Delta \vec{\Phi}_i[\vec{x}_i, t+1] = T_i \vec{\Phi}_i[\vec{x}_i, t], \tag{7}$$

where $\Delta \vec{\Phi}_i$ is the change of the distribution during one time step. In this case the distribution at a given time is represented by the vector $\vec{\Phi}_i[\vec{x}_i, t]$, whose entries correspond to the probability densities of $x_i \in \{0, 1/N, 2/N, \ldots, 1\}$. Upon multiplying by the matrix of transition probabilities, $T_i$, the time change of the distribution is obtained. Because only transitions are considered, the main diagonal of $T_i$ equals zero, while the upper and lower diagonals equal Eqs (1b) and (1c), respectively. Host death is reflected in additional non-zero probabilities, $\tau$, at the first column for microbial taxa ($i \geq 1$) or last column for unoccupied space ($i = 0$). The non-trivial stationary distribution $\vec{\Phi}_i[\vec{x}_i]$ occurs for $\Delta \vec{\Phi}_i[\vec{x}_i, t+1] = \vec{0}$, corresponding to the eigenvector of $T_i$ with eigenvalue zero. We used this method to compute the stationary distribution in Python 3.6.

If numerical problems emerged solving Eq (7), we focused on solving $\vec{\Phi}_i[\vec{x}_i, t+1] = R_i \vec{\Phi}_i[\vec{x}_i, t]$ for $\vec{\Phi}_i[\vec{x}_i, t+1] = \vec{\Phi}_i[\vec{x}_i, t]$ instead. Here $R_i$, the probability matrix, is identical to $T_i$, except at the main diagonal where it equals Eq (1d). The stationary distribution corresponds to the eigenvector of $R_i$ with eigenvalue one.

## Stochastic simulations

To study the transient dynamics of colonization and test our stationary estimation, we performed individual-based simulations. These were performed for 500 hosts, $N = 10^4$, two equally abundant microbial taxa in the pool of colonizers, $p_1 = p_2 = 0.5$, and initially sterile hosts ($x_0 = 1$ and $x_1 = x_2 = 0$ as initial condition). We varied the values of migration ($m$) and rate of occupation of empty space ($\alpha_0$).

## Difference between models

To compare models considering finite ($\tau > 0$) and infinite host lifespan ($\tau = 0$), we calculated the total difference between their stationary distributions, $\Phi_i[x_i]|_{\tau > 0}$ and $\Phi_i[x_i]|_{\tau = 0}$, for all $x_i$

$$\frac{1}{2} \sum_{x_i} |\Phi_i[x_i]|_{\tau > 0} - \Phi_i[x_i]|_{\tau = 0}| \tag{8}$$

This difference, ranging from 0 to 1, will equal zero only if for all $x_i$, the two distributions are identical, $\Phi_i[x_i]|_{\tau > 0} = \Phi_i[x_i]|_{\tau = 0}$.

## Probability of microbe-free, colonized and fully-colonized hosts

To analyse when a particular microbial taxon will not be observed in a host, i.e. its probability of non-colonization, we calculated

$$P\left[x_i < \frac{1}{N}\right] = \Phi_i[0], \tag{9}$$

where $1/N$ is the minimum observation limit and $P[x_i \geq 1/N] = 1 - P[x_i < 1/N]$ is the probability of colonization by microbe $i$.

On the other hand, to analyse when a particular microbial taxon will fully occupy a host, we calculated

$$P\left[x_i > \frac{N-1}{N}\right] = \Phi_i[1], \tag{10}$$

where $\frac{N-1}{N}$ is the maximum observation limit, and $P[x_i \leq (N-1)/N] = 1 - P[x_i > (N-1)/N]$ is the combined probability of partial and non-colonization.

Finally, the quantities $P[x_0 < 1/N]$ and $P[x_0 > (N-1)/N]$, indicate the probability of hosts full of microbes and the probability of hosts free of microbes, respectively.

## Alternative microbiome states

To assess the modality of the distribution $\Phi_i[x_i]$, i.e. alternative microbiome states, we identified the maxima of the distribution of its numerical solution for varying parameters. The distribution can be unimodal, with the maximum located at one of the boundaries or between them, $x_i = \{0, x_i^*, 1\}$, or bimodal, by a combination of the former. We classified these states and calculated the magnitude of their maxima.

## Comparison between the model and simulated data

In order to evaluate our model, we compared it to stochastic simulations (S1, S2 and S3 Figs). As mentioned above, we simulated hosts individually. However, our model provides a population description for overlapping generations. Therefore, we sampled single time steps of the colonization trajectories according to Eq (6), which indicates the probability of a host death-birth event through time. The distribution of the simulated sampled set was then compared to our theoretical model predictions.

## Code availibility

The Python code for simulations, numerical solution of the model and figures is available at https://github.com/romanzapien/microbiome-hostspan.

## Results

### The dynamics of colonization affects the microbiome of finite-living hosts, but not of infinite-living habitats

The formation of a microbiome goes through several stages. Analytically, much of the focus has been on its long-term equilibrium, assuming hosts with infinite lifespan. Much less is known about the transient stage. Fig 2 shows two illustrative individual-based simulations, where hosts are colonized by two neutral microbial taxa, going from a microbe-free to a microbe-occupied state. The dynamics is qualitatively different depending on $\alpha_0$: For $\alpha_0 = 0$, the host is colonized by the two microbes at the same time, leading to a unimodal distribution

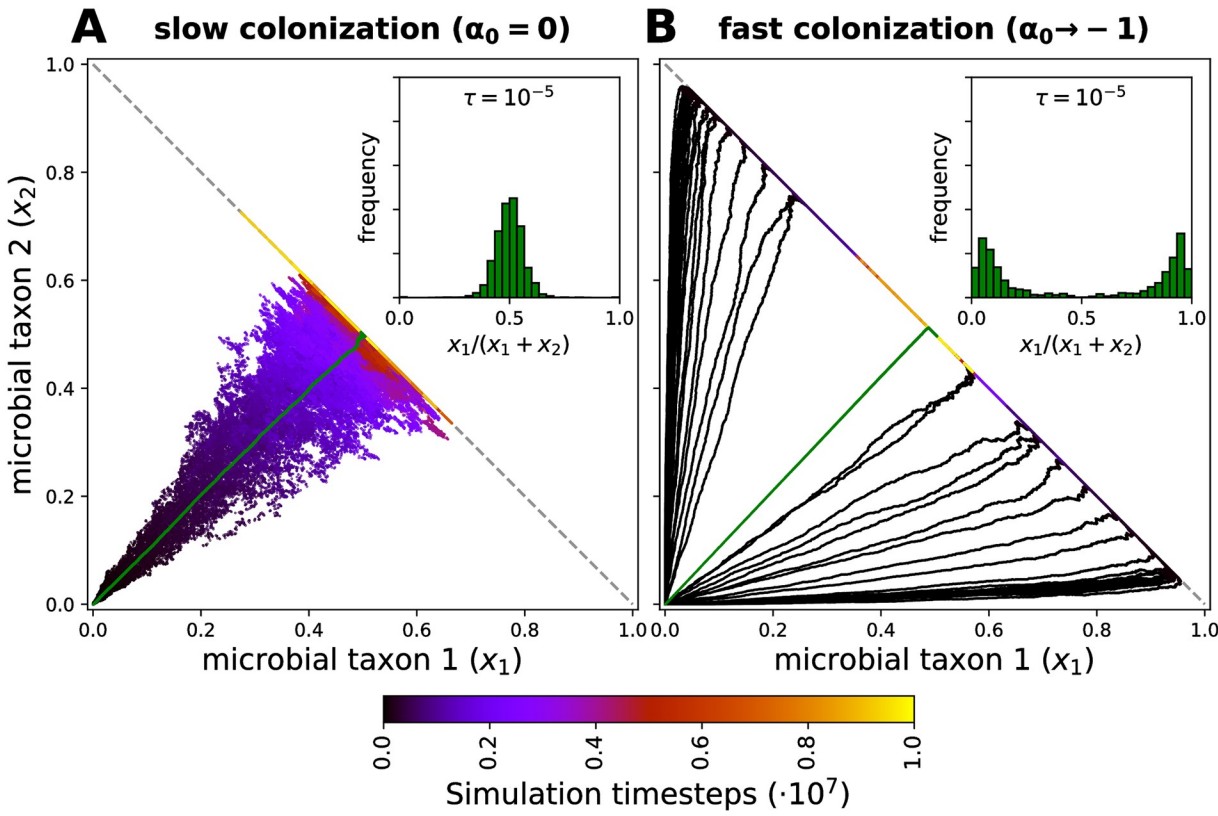

**Fig 2. Individual-based simulations of colonization for two neutral microbial taxa.** The colonization trajectory of 50 microbe-free hosts is shown, with colors indicating the time of sampling. Each trajectory is composed of $10^4$ points, with the green line indicating the mean and the dashed line indicating full colonization. Insets show the distribution of $x_1/(x_1 + x_2)$ for time steps sampled according to a host death-birth probability $\tau = 10^{-5}$ in Eq (6). (**A**) When hosts are colonized slowly, the trajectories maintain a mean frequency given by the pool of colonizers ($p_i$), but show an increased standard deviation before full colonization, which decreases later to reach $x_i \approx p_i$ at the long-term equilibrium. (**B**) When hosts are colonized rapidly, the mean frequency across many hosts is conserved, but the distribution becomes bimodal as a result of the fast proliferation of the first colonizer, and a slower convergence to the long-term equilibrium. The inset shows the distribution of taxon 1 over the simulated time. For finite host lifespans and fast colonization, such dynamics can produce alternative microbiome states at the population level (inset in B). Sampling was performed every 10 time steps in a simulation of $10^7$ time steps. Other parameters: $N = 10^4$, $m = 0.01$, $p_1 = p_2 = 0.5$, and $\alpha_1 = \alpha_2 = 0$.

that is similar to the long term equilibrium even during the transient. For $\alpha_0 < 0$ empty space is occupied more rapidly compared to the dynamics between microbes. This leads to a situation where one microbial strain dominates the host until the host is fully colonized, leading to a bimodal distribution in the colonization of hosts. Only on a much longer timescale, this distribution is replaced by the unimodal distribution characteristic for the long term equilibrium.

Given a low rate of external colonization ($m \to 0$), the time required for full colonization will be shorter than that to reach the long-term equilibrium. Such difference will increase even further for rapid colonization, $\alpha_0 < 0$. When considering a finite host lifespan ($\tau > 0$), this difference in time-scales will influence the expected microbiome composition. Interestingly, for shorter lifespans, the host population might be multimodal and only partially colonized (Fig 2B). Moreover, for sufficiently small external colonization and short host lifespan, coexistence of colonized and microbe-free individuals is expected (S4 Fig).

From a microbial point of view, the results shown here occur in a completely neutral context. They can also be generalized to cases with many microbial taxa. A non-neutral dynamics of the microbes ($\alpha_i \neq 0$) will modify the stationary distribution, i.e. they will not only depend

on the frequency in the pool of colonizers ($p_i$) and host lifespan (via $\tau$). Instead, asymmetries of the multimodality and differential colonization are expected once $\alpha_i \neq 0$ is assumed.

## A short host lifespan influences the microbiome

We quantified the change of the stationary distribution caused by a finite host lifespan systematically. Using the stationary distribution of the frequency, $\Phi_i[x_i]$, we compared the predictions assuming hosts with infinite lifespan ($\tau = 0$) against those with hosts with finite lifespan ($\tau > 0$). Such comparisons were done for multiple migration probabilities ($m$), frequencies in the pool of colonizers ($p_i$), and rates of empty space occupation ($\alpha_0$). As explained in the Methods, Eq (8), we express the results as the difference between the stationary distributions.

Figs 3 and 4 show the results of the microbial load (total microbial frequency) and frequency of a particular microbe, respectively. Within the range of $m$ and $\tau$ analysed, the difference is always greater than zero, indicating the importance of $\tau$ in our model and the predictions arising from it. Only for $\tau \rightarrow 0$, full agreement is expected.

Regarding the microbial load, infinitely living hosts ($\tau = 0$) provide enough time for them to be fully colonized and for the distribution of microbes to reach an equilibrium. In contrast, a finite lifespan ($\tau > 0$) might not allow full colonization before host death. For a slow occupation of empty space ($\alpha_0 = 0$) the difference increases with shorter lifespan (large $\tau$) and reduced migration (small $m$), Fig 3A. In this case, the model with $\tau = 0$ predicts a distribution centered at frequency 1 decaying towards 0, while the model with $\tau > 0$ predicts a sharp maximum centered at frequency 0 decaying towards 1. In contrast, rapid occupation of empty space ($\alpha_0 < 0$) causes the difference to decrease and to become increasingly independent of $m$ (Fig 3B). This occurs because the time for colonization, i.e. host lifespan, becomes more relevant than migration, as successful migrants are increasingly likely to proliferate within hosts.

For a specific microbial taxon, infinitely living hosts ($\tau = 0$) allow the frequency in the hosts to reach that in the pool of colonizing microbes ($p_i$). However, a restricted, finite lifespan ($\tau > 0$) might not allow to reach this value. In our model, the relevance of $\tau$ increases with its magnitude, but not independently of $m$. The maximum difference between the two distributions occurs for short lifespan (large $\tau$) and large migration (larger $m$) as $p_i \rightarrow 0$ (Fig 4B and 4C). In this region, the model with $\tau = 0$ predicts a distribution centered at $x_i \approx p_i$, while the model with $\tau > 0$ predicts a distribution centered at $x_i = 0$ decaying towards 1. Finally, for a single colonizing taxon ($p_i = 1$, Fig 4A) the difference increases analogously to Fig 3A, i.e. the difference increases for smaller migration and shorter lifespan.

## Microbe-free, colonized hosts, and their coexistence are expected

A major consequence of a host finite lifespan is the coexistence of hosts with various degrees of colonization, including microbe-free hosts. We calculated the probability of full colonization in the stationary distribution, i.e. $P[x_0 < 1/N]$ (Eq (9)), for different parameters given a certain capacity for microbes ($N$).

Fig 5 shows the effect of $m$, $\tau$, and $\alpha_0$ on the probability of full colonization. Different parameter combinations can result in the same probability of full colonization. Partial colonization is the most likely state for short host lifespans (large $\tau$). Only for long living hosts (small $\tau$), both death probability $\tau$ and migration $m$ are important, with $m$ having a larger impact on the distribution when it is larger (Fig 5A). Finally, a faster occupation of empty space ($\alpha_0 < 0$) makes the probability of full colonization less dependent on $m$ and increases it for shorter living hosts (larger $\tau$), i.e. the coexistence with partially colonized hosts becomes less likely (Fig 5B).

## Difference between distributions

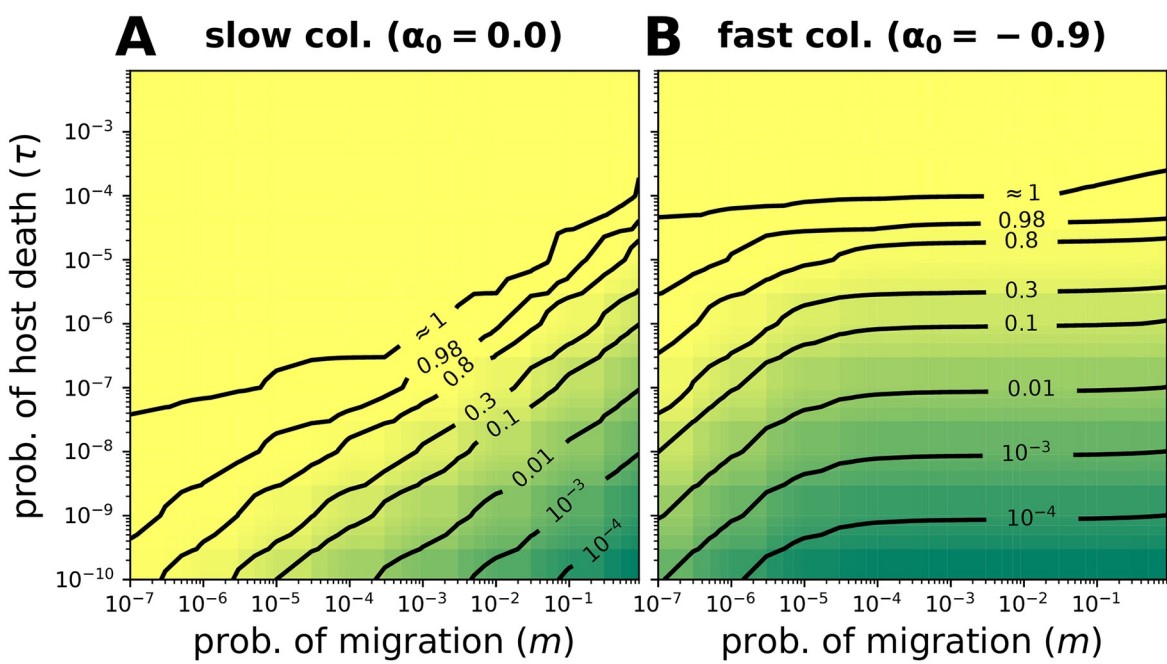

## Shape of stationary distribution

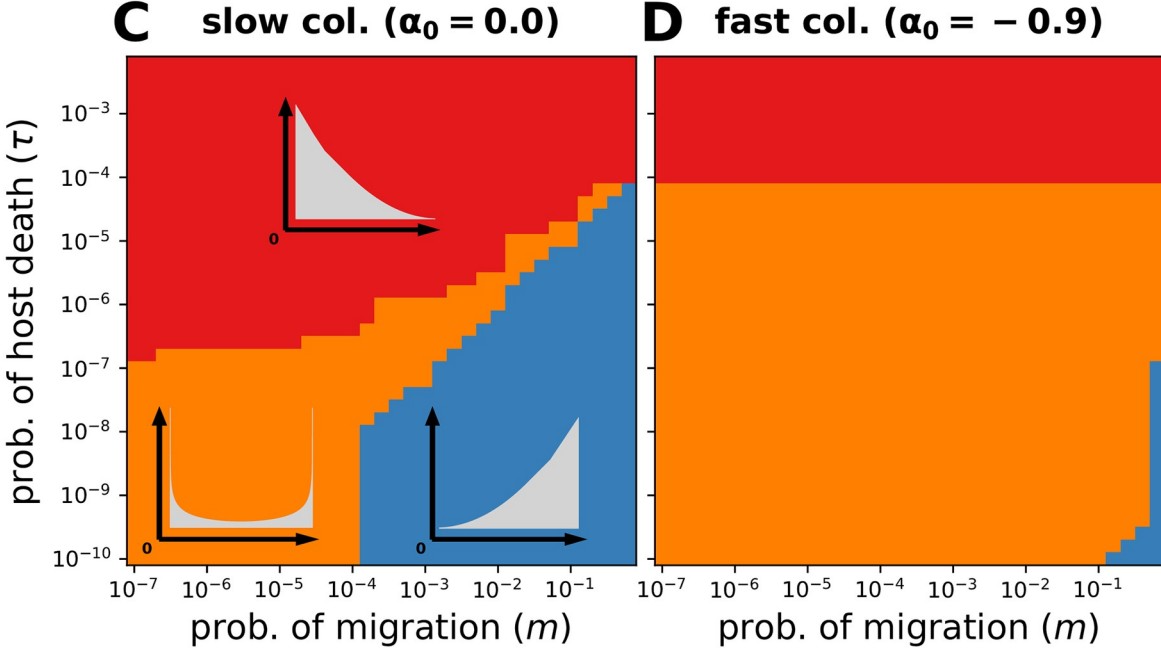

**Fig 3. Microbial load.** (**A**-**B**) The difference between models with finite ($\tau > 0$) and infinite ($\tau = 0$) host lifespan is shown, Eq (8). (**A**) For a slow occupation of empty space, the difference is maximal for small migration ($m$) and large $\tau$ as the model with $\tau = 0$ predicts a distribution centred at frequency 1 decaying towards 0, whereas the model with $\tau > 0$ predicts a distribution centred at frequency 0 decaying towards 1. For a fixed $\tau$ the difference is always greater for smaller $m$. Only for $\tau \gtrsim 10^{-4}$ the difference is maximal and independent of $m$. Finally, a smaller $\tau$ always approximates the models; nonetheless within the range analysed the difference is always greater than zero. (**B**) A faster occupation of empty space decreases the difference and makes it increasingly independent of $m$, as $\tau$ dominates the predictions of the model. (**C**-**D**) The distributions are classified according to their number of maxima (unimodal or bimodal) and location (0 and 1). (**C**) A slow occupation of empty space results in microbe-free hosts being the maximum for short host lifespans (large $\tau$), fully colonized hosts for large migration ($m$) and small $\tau$, or microbe-free and microbe-occupied hosts simultaneously

for small $m$ and $\tau$. The bimodality results from a limited migration preventing all the hosts from being colonized but over a host lifespan sufficient for successful colonizers to occupy host fully. (**D**) A faster occupation of empty space increases the bimodality region at the expense of the unimodal cases. In this case, $\alpha_0 \rightarrow -1$ favours the microbe-occupied maximum. When classifying the distributions, any probability smaller than $10^{-9}$ was considered as zero. Other parameters: $N = 10^4$. We use Eq (5a) where no definition of $p_i$ and $\alpha_i$ is required.

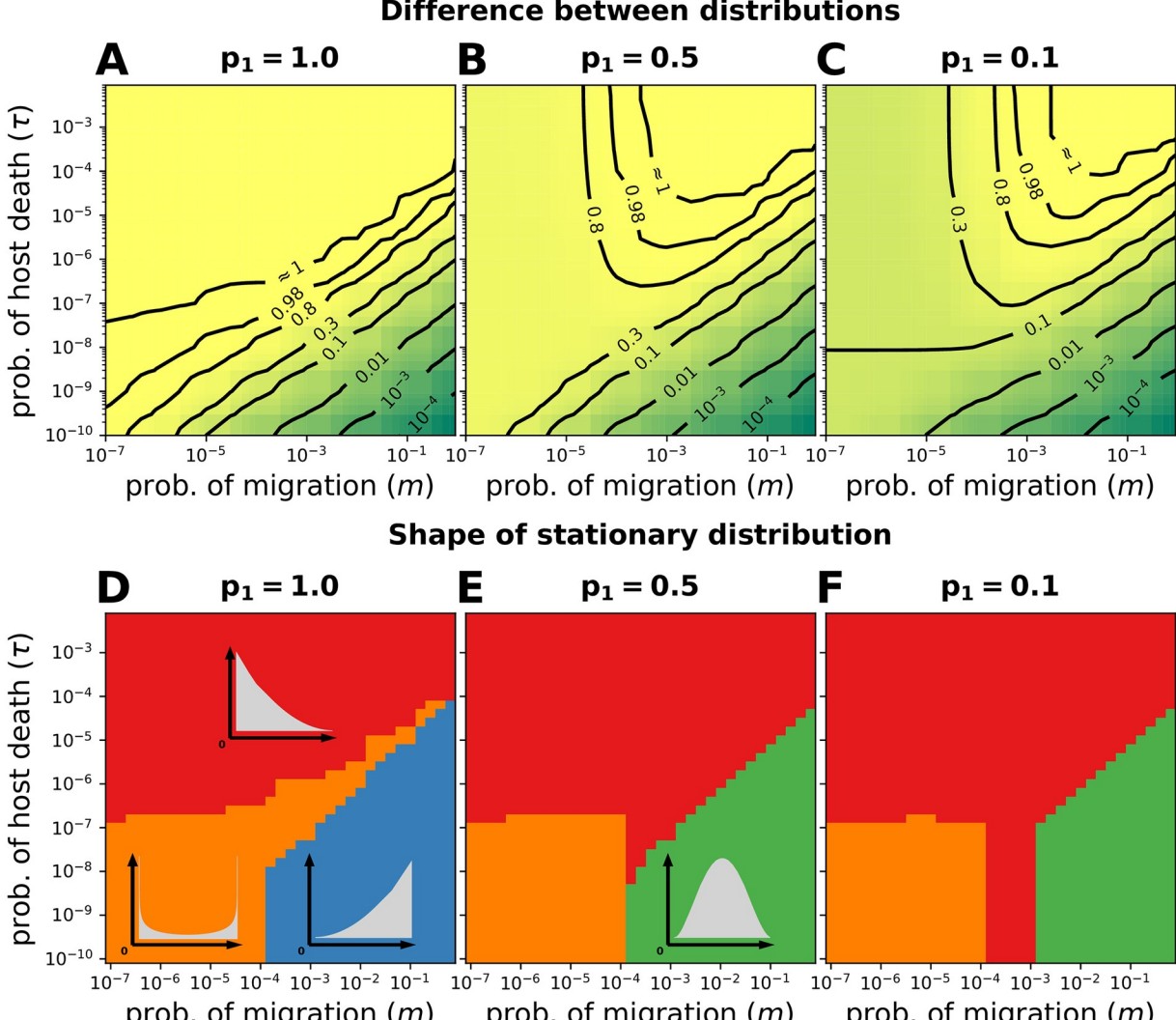

**Fig 4. Microbial taxon 1.** $p_1$ indicates the frequency of microbial taxon 1 in the pool of colonizers. (**A-C**) The difference between models with finite ($\tau > 0$) and infinite ($\tau = 0$) host lifespan is shown, Eq (8). (**A**) A single colonizing taxon follows the same pattern shown in Fig 3A. (**B-C**) The maximal difference of a less abundant colonizing microbe changes in the direction of larger $m$. (**D-F**) The distributions are classified according to their number of maxima (unimodal or bimodal) and location (0, 1, and an internal maximum). (**D**) A single colonizing taxon mirrors Fig 3C, and bimodality is prevalent. (**E-F**) Less abundant taxa have a decreased probability of colonization, and an internal maximum emerges for large $m$ and long host lifespan (small $\tau$), whose location is influenced by the frequency in the pool of colonizers ($p_1$). When classifying the distributions, any probability smaller than $10^{-9}$ was considered as zero (Other parameters $N = 10^4$ and $\alpha_0 = \alpha_1 = 0$). S7 Fig shows how the frequency $x_1$ changes as we increase $\tau$ for $m = 10^{-3}$.

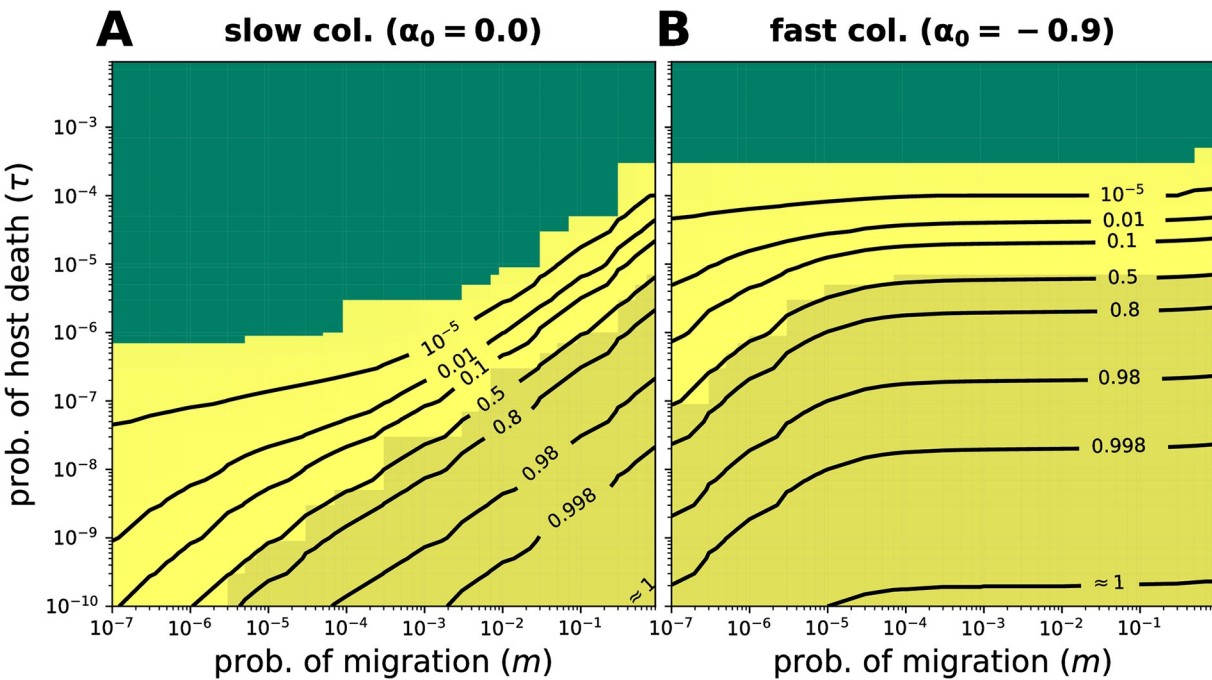

**Fig 5. Probability of full colonization in the stationary distribution.** The probability of full colonization $P[x_0 < 1/N]$ is shown, Eq (9), where $1/N$ is the minimum observation limit. For short host lifespans (large $\tau$), partial colonization is more common than full colonization. (**A**) For long host lifespans (small $\tau$) or large migration ($m$), $m$ has an effect on the probability, but this is lost as $\tau$ is larger and $m$ smaller. (**B**) A faster occupation of empty space increases the probability of full colonization, but migration ($m$) influence is now restricted to long host lifespans (small $\tau$) and small $m$. I.e. the host lifespan ($\tau$) becomes the most relevant parameter. Other parameters: $N = 10^4$. We use Eq (5a) where no definition of $p_i$ and $\alpha_i$ is required.

The results shown in Fig 5 depend heavily on the capacity for microbes of the host ($N$). Decreasing $N$ causes the hosts to be fully colonized quicker; thus partially colonized hosts will be observed for shorter host lifespans (larger $\tau$), slower occupation of empty space (larger $\alpha_0$), and less migration (smaller $m$), S5 Fig. The opposite is expected for larger $N$.

As shown by our calculations, S6 Fig, we argue that even microbe-free hosts might not be an experimental artefact, but an inherent outcome of the host colonization process in some host-microbiome systems [23, 24], even under neutral (i.e. non-selective) conditions [8]. This might be evident for short living hosts, but less so for longer lifespans. In such case, its experimental observation might be possible only for large samples of hosts.

### Rapid proliferation of the first colonizer can result in alternative microbiome states

We have noted previously the existence of multimodal distributions in the transient colonization, and how these prevail in the stationary distribution due to the finite lifespan of hosts (Fig 2). A particular microbial taxon might either succeed or fail to colonize a host, leading to the coexistence of hosts with alternative microbiome states. Moreover, in specific cases all possible microbes could succeed or fail to colonize a host, allowing the coexistence of microbe-free and microbe-occupied hosts. These extremes can have similar or different magnitudes, as shown in Fig 5 and S6 Fig.

Fig 3C and 3D shows the stationary distribution of microbial load for different rates of empty space occupation, $\alpha_0$. Firstly, a large host death-birth probability ($\tau$) causes hosts to be

rarely colonized; hence most remain microbe-free, so $x_0 = 1$ is the only maximum. Secondly, a large migration ($m$) and small $\tau$ provides enough time for hosts to be fully colonized, so $x_0 = 0$ is the only maximum. Finally, the processes of limited migration and long host lifespan combine to define a region where bimodality is expected (Fig 3C). The magnitude of the maxima and region of bimodality are influenced by $\alpha_0$ (Fig 3D), with $\alpha_0 \to -1$ favouring the microbe-occupied over the microbe-free state (Fig 5 and S6 Fig).

Similarly, Fig 4D–4F shows the stationary distribution for various frequencies of a microbial taxon in the pool of colonizers ($p_i$) and $\alpha_0 = 0$. A qualitative description of the complete distributions (see S7 Fig) is shown. Again, bimodality is expected for small $m$ and large $\tau$. Many microbes do not colonize, but successful colonizers proliferate to occupy hosts entirely. The bimodality region is shaped by $p_i$. A single colonizer ($p_i = 1$, Fig 4D) mirrors Fig 3C. In contrast, $p_i < 1$ has the effect of vanishing the bimodality if $m$ or $\tau$ are larger (Fig 4E and 4F). Outside this region, a large $\tau$ causes most hosts to be microbe-free, so $x_i = 0$ is the only maximum. However, a larger $m$ and smaller $\tau$ make $x_i = 1$ the single maximum if $p_i = 1$, or an internal maximum if $p_i < 1$. Finally, the split into alternative states might be reinforced if empty space is occupied more rapidly, $\alpha_0 < 0$ (Fig 2 and S4 Fig). This results from a limited migration and rapid proliferation of the first colonizer. Although the alternative states could be transient for long-living hosts, they might persist for short-living ones.

## By reducing the colonization probability, the finite host lifespan makes the core microbiome context-dependent

Previous research has focused on defining the set of microbial taxa consistently observed in a given host species. This is often called the core microbiome. In our model, stochastic colonization reduces the probability of observing a taxon in all hosts (Fig 6). Importantly, this is not caused by any kind of selection or competition, but by migration ($m$), time for colonization (via $\tau$), capacity for microbes ($N$), and the frequency of a colonizing taxon ($p_i$) alone. Fig 6 shows the probability of observing a microbial taxon within a host, $P[x_i \geq 1/N]$, for different values of $m$, $\tau$, and a fixed $N$. For the values of $p_i$ shown, the contour lines increasingly depend on $\tau$ for larger $\tau$. Successful colonization is more prevalent whenever $m$ is larger and $\tau$ smaller, for microbes down to a frequency of $p_i = 0.1$. Nonetheless, even a single colonizing taxon could not consistently be observed for some $m$ and $\tau$ (Fig 6A and S8 Fig). Finally, a smaller microbial frequency in the pool of colonizers ($p_i$) reduces the overall colonization probability (Fig 6B and 6C, smaller values are shown in S8 Fig).

These results suggest that under neutral dynamics, the observed frequency of microbes within hosts, i.e. the colonization probability, cannot be universally used to define a core microbiome, as the frequency of readily colonizing taxa depends on host and microbial features.

## Discussion

Although microbes are ubiquitous in nature [25], including the human body [26], it remains to be answered which microbes not only transit from the environment to the hosts but also persist in or on them. Our understanding of these processes relies on identifying the factors underlying host colonization.

We have introduced a stochastic model along the lines suggested by [27], where migration and death-birth processes of microbes within hosts with finite lifespans can produce a range of colonization dynamics and distinctly different microbiomes—even when there is no selection at all (Fig 1). A key assumption in our model is the absence of inheritance of microbes [28], as hosts are colonized after birth from the environment only. In this context, the microbiome is

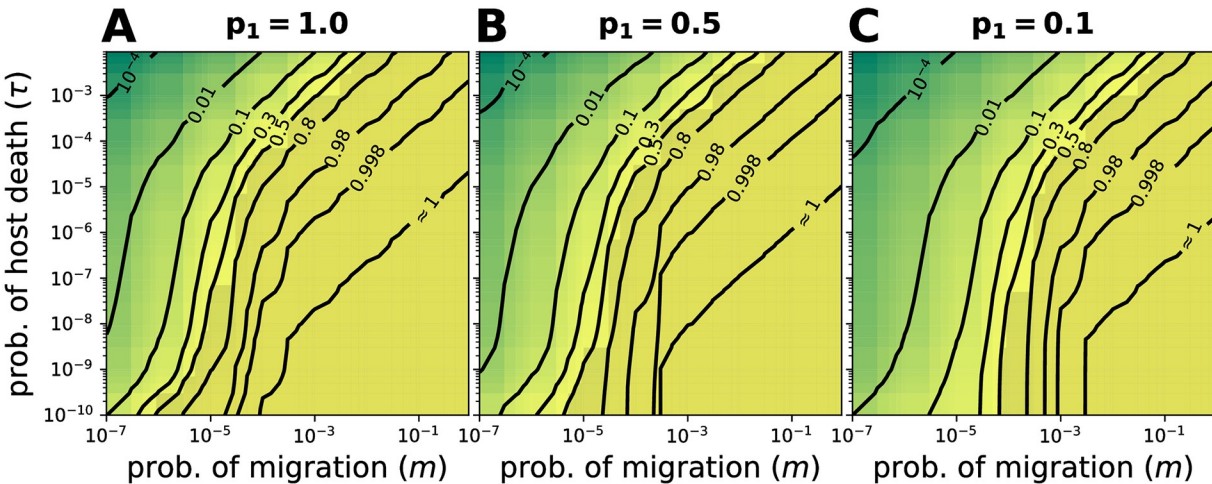

**Fig 6. Probability of colonization of microbial taxon 1 in the stationary distribution.** $p_1$ indicates the frequency of microbial taxon 1 in the pool of colonizers. The probability that a particular microbe is present, $P[x_1 \geq 1/N]$, is shown, where $1/N$ is the minimum observation limit. (**A**) A single microbial taxon colonizes for a large combination of migration ($m$) and probability of host death-birth ($\tau$). The probability increases with $m$ and with longer lifespan (small $\tau$). (**B-C**) For less abundant colonizing microbes, the probability is reduced. S8 Fig shows the effect of $p_1$ on $P[x_1 \geq 1/N]$. Other parameters: $N = 10^4$ and $\alpha_0 = \alpha_1 = 0$.

driven by the frequency in the pool of colonizers. This frequency (which is constant in our model) does not need to be the frequency of an environmental microbe, but can more generally be a function of it. Several organisms including *D. rerio* [29], *C. elegans* [8], and *D. melanogaster* [24] might be colonized from the environment only. Others have weak inheritance [30], or might be microbe-free prior to birth, like humans [15]. Many host species will also inherit their microbes from their parents.

Critical to colonization in our model is the magnitude of the microbial migration from the environment to the hosts ($m$) [27]. As observed in the gut of *D. rerio* [31], microbial migration could overwhelm other host selective and non-selective processes. In addition, we have combined the host lifespan with a constant microbial cell doubling time [32] to define $\tau$ as the parameter of timescale separation between hosts and microbes. This serves as an indicator of the relevance of a host population dynamics for the microbiome dynamics. In agreement with [29], we observe that a limited migration imposes a bottleneck on the colonizers, which combined with a finite host lifespan might produce complicated colonization patterns (Fig 2 and S4 Fig). The parameters $m$ and $\tau$ have allowed us not only to classify the stationary colonization distributions (Figs 3 and 4), but also to quantify the relevance of the finite host lifespan in our model (Figs 3 and 4).

The parameters $m$ and $\tau$ can be inferred from data. Alternatively, prior knowledge of the host lifestyle can give us intuition. For example, given the short lifespan of *C. elegans* a large $\tau$ is expected; while its feeding mechanism might pose a bottleneck, suggesting a small $m$. In principle, $m$ can range from 0 (no environmental microbes going in) to 1 (only external migration and no internal reproduction). This range is spanned by previous studies that estimated this parameter for multiple species [5–7].

Sloan et al. [3] developed a neutral model to estimate the equilibrium distribution of a microbiome in an infinite-living habitat. Several studies have fit this model to data of different host species [5–7]. However, based on our results for hosts with varying lifespans, we predict that Sloan et al.'s model will perform poorly for hosts with short lifespans, e.g. *D. rerio*, *D.*

*melanogaster*, and *C. elegans*, impeding comparisons of neutrality between host species (Figs 3 and 4). On top of that, the average microbiome of all sampled hosts might be a transient state, not the long-term equilibrium that is assumed when fitting the model. These problems are expected to be even more pronounced for low frequency microbial taxa (Fig 6 and S8 Fig), and small host populations samples.

As going from a microbe-free to a colonized state might affect the expected stationary distribution in hosts with finite lifespans, we included the occupation of empty space by microbes in our model explicitly. Then, we computed the probabilities of observing microbe-free (S6 Fig), fully colonized hosts (Fig 5), and their coexistence (Fig 3). Interestingly, there is building evidence of individuals with microbe-free guts coexisting in *D. melanogaster* [24], *C. elegans* [8], and caterpillars where a microbe-free state might be prevalent [23]—supporting our results. We argue that in such host species, both a low microbial migration and short host lifespan might be causative [33].

We have also observed alternative microbiome states. In other words, subsets of hosts whose microbiome is dominated by different microbial taxa (Fig 2). Our results suggest this might occur for low microbial migration and short host lifespan (Fig 4). Recently, [8] have observed alternative microbiome states occurring in *C. elegans* when this is colonized by two neutral *Escherichia coli* strains. The implications of our results go beyond colonization, as they predict priority effects [34], life history [35], and timing to be important conditions for any host control mechanism. Furthermore, we provide a generative process for the emergence of different microbiome states in the gut [36], that does not rely on selection, interaction networks or environmental change [37, 38]. Our results support the current view that the enterotypes often discussed are indeed states contained in a continuum of colonization [39].

Finally, we have addressed the issue of identifying a core microbiome. In contrast to the present interest on identifying this subset of microbes [40], we argue that intrinsic features of the colonization process might impede finding a consistent subset. Specifically if the observed frequency within hosts is the criterion (Fig 6). More informative, however, would be distinguishing potential from factual colonizers, with members of the latter depending on the context where the colonization happens. We stress the relevance of regarding the colonization and coexistence ahead of the coevolution of hosts and microbes. Let alone, their organismic nature and implications [41, 42].

As a consequence of the neutral assumption (fitness $\alpha_i = 0$ in Eq (1) for $i \geq 1$), our results extend to microbiomes with an arbitrary number of taxa. Although we first illustrate the process with two of them (Fig 2), analogously to [8], we move on to focus on the perspective of a single taxon ($x_i$ in Eqs (3b), (4b) and (5b)). In this view, the collection of other taxa can be arbitrarily complicated. This is particularly important in conditions leading to alternative microbiome states, where the frequency in the pool of colonizers, $p_i$, becomes extremely relevant. While symmetric $p_i$ across taxa will result in as many alternative states as taxa, asymmetries will make those with larger $p_i$ appear more prominent, giving the impression of a reduced number of alternative states [39].

Future empirical work could focus on characterizing the prevalence of effects associated with the short lifespan—slow immigration regime (Fig 2). Although this depends on the timescale of the microbial dynamics also (resulting from the quality of the host as a habitat), host life-history might provide direction (Fig 1B). For example, a short lifespan together with a reduced amount of microbes reaching the gut, indicate the potential of observing such regime in nematodes [8] and some insects [23, 24, 33]. Moreover, different tissues within a host might provide different conditions. Other hosts might be subtler. As our model indicates, different life-histories might lead to similar results (contours in Figs 3–6).

We have presented a minimal neutral model. More complex processes could build upon it. Among others, the influence of the prenatal microbiome on the dynamics and stationary distribution in a neutral context is largely unknown [9–11]. Additionally, after an initial stochastic assembly, hosts might actively influence their microbiome via immunity and development [14]. This might have general or taxa specific effects. Particularly relevant as well, is the role that first colonizers (Fig 2) might have in modifying the internal host, influencing the arrival of upcoming microbes [43]. This could reinforce the difference between alternative microbiome states, at taxonomic and functional levels. Finally, as reported in some hosts [44], non-smooth changes of the microbiome could occur. These changes, of intrinsic (e.g. microbial succession [43], host and metabolic rhythms [45]) or extrinsic (e.g. diet change [43], disease, and antibiotics [46]) origin might be more akin to a Lévy walk [16].

Although previous models have studied signatures of ecological neutrality and selection in microbiome data [47, 48], as well as its evolution [9, 49], they have not described the ecological effects that we have described here. We share Roughgarden et al.'s [50] view that an eco-evolutionary approach is needed, but our results emphasize that colonization in a neutral context might already be sufficient to unify important and disconnected experimental observations, often implicitly attributed to selection. Non-neutral processes might then build on top of such patterns.

## Conclusion

We have introduced a stochastic model of the colonization of microbe-free hosts. After considering the environmental colonization and finite lifespan of hosts, our model recapitulates patterns reported experimentally. Namely, the coexistence of microbe-free and partially colonized hosts, as well as alternative microbiome states; both depending especially on the host lifespan. Crucially, our observations occur under non-selective conditions at the level of microbes or hosts. The model and results presented here aim to provide a null model for studying the host-microbiome formation by assuming the neutrality of microbial taxa—without ruling out that also selection will be important for these processes in nature. But even in the absence of any selective differences, our model explains a wide range of recent observations in microbiomes, from the observation of non-colonized hosts to alternative microbiome states.

## Supporting information

**S1 Appendix. Derivation of the Fokker-Planck approximation with resetting.**
(PDF)

**S1 Fig. Comparison between simulations and the model: Probability of microbe-free hosts in the stationary distribution.** The $P[x_0 > (N − 1)/N]$ is shown, Eq (10). Lines show the model prediction, while triangles show the average over the steady state of 500 host samples according to Eq (6). The match spans several magnitude orders of migration ($m$) and probability of host death-birth events ($\tau$). The probability increases for shorter host lifespans (larger $\tau$) and less migration to the hosts (smaller $m$). The rate of occupation of empty space ($\alpha_0$) has a larger effect on cases where migration is limited and the host lifespan is long (small $\tau$). Simulations were computed as explained in the Methods. Other parameters: $N = 10^4$. We use Eq (5a) where no definition of $p_i$ and $\alpha_i$ is required.
(TIF)

**S2 Fig. Comparison between simulations and the model: Probability of full colonization in the stationary distribution.** The $P[x_0 < 1/N]$ is shown, Eq (9). Lines show the model prediction, while triangles show the average over the steady state of 500 host samples according to Eq

(6). The match spans several magnitude orders of migration ($m$) and probability of host death-birth events ($\tau$). The probability increases for longer host lifespans (smaller $\tau$) and larger migration to the hosts (larger $m$). The rate of occupation of empty space ($\alpha_0$) has a larger effect on cases where migration is large and the host lifespan is long (small $\tau$). Simulations were computed as explained in the Methods. Other parameters: $N = 10^4$. We use Eq (5a) where no definition of $p_i$ and $\alpha_i$ is required.
(TIF)

**S3 Fig. Comparison between simulations and the model: Probability of colonization of microbial taxon 1 in the stationary distribution.** $p_1$ indicates the frequency of microbial taxon 1 in the pool of colonizers. Lines show the model prediction, while triangles show the average over the steady state of 500 host samples according to Eq (6). The match spans several magnitude orders of migration ($m$) and probability of host death-birth events ($\tau$). The probability increases for longer host lifespans (smaller $\tau$) and larger migration to the hosts (larger $m$). Simulations were computed as explained in the Methods. Other parameters: $N = 10^4$ and $\alpha_0 = \alpha_1 = 0$.
(TIF)

**S4 Fig. Individual-based simulations of colonization for two neutral microbial taxa: Limited migration.** Except for $m = 10^{-4}$, all parameters are equal to those in Fig 2. (**A**) The limited migration and slow empty space occupation impedes the host from being colonized completely. (**B**) When empty space is occupied rapidly, although the mean is conserved, the distribution becomes sharply bimodal as a result of the fast proliferation of the first colonizer, and a slow convergence to the long-term equilibrium, which within the time-range simulated is not reached. For finite host lifespans, such dynamics can produce alternative microbiomes and partial colonization of hosts in the equilibrium.
(TIF)

**S5 Fig. Probability of full colonization in the stationary distribution: Smaller microbiome size.** Except for $N = 10^3$, all parameters are equal to those in Fig 5. The smaller capacity for microbes of a host makes full colonization more likely, and migration ($m$) has increased influence for larger $\tau$.
(TIF)

**S6 Fig. Probability of microbe-free hosts in the stationary distribution.** The $P[x_0 > (N - 1)/N]$ is shown, Eq (10). (**A**) Migration ($m$) is the main driver of the microbe-free state, but still interacting with the probability of host death-birth ($\tau$). The microbe-free state prevails for small $m$, increasing in the direction of a short host lifespan (large $\tau$). (**B**) Although a faster occupation of empty space decreases its probability, microbe-free hosts are still expected. Moreover the host lifespan (via $\tau$) becomes more relevant. Other parameters: $N = 10^4$. We use Eq (5a) where no definition of $p_i$ and $\alpha_i$ is required.
(TIF)

**S7 Fig. Probability density of microbial taxon 1 as a function of host death.** The cross-section of Fig 4 where $m = 10^{-3}$ is shown. (**A**). If there are only microbes of type 1 in the pool of colonizers ($p_1 = 1$), small $\tau$ implies that there is a single maximum at $x_1 = 1$—the hosts tend to be fully occupied. Bimodality is observed for $7 \cdot 10^{-8} \lesssim \tau \lesssim 10^{-6}$—some hosts are occupied, but some remain empty. For large $\tau$, hosts tend to remain empty and the distribution has a single maximum at $x_1 = 0$. Black lines indicate the boundaries separating them (see Fig 4). (**B**) If the microbe is present in the pool of colonizers at $p_1 = 0.5$, no bimodality is observed. For small $\tau$ the frequencies are representative of the pool of colonizers and for large $\tau$ most hosts

do not contain microbe 1. (**C**) If the microbe is rare in the pool of colonizers, $p_1 = 0.1$, the distribution has a single peak at $x_1 = 0$. This occurs for all values of $\tau$ shown here, because there is not enough time in the host to reflect the small number of microbe 1 individuals in the pool of colonizers (any probability smaller than $10^{-9}$ was considered as zero, $N = 10^4$ and $\alpha_0 = \alpha_1 = 0$).
(TIF)

**S8 Fig. Probability of colonization of microbial taxon 1 as a function of its frequency in the pool of colonizers.** The results of multiple probabilities of host death-birth events ($\tau$) are shown. Overall, the probability of colonization increases with the frequency in the pool of colonizers ($p_1$), but decreases as the host lifespan shortens (larger $\tau$). A smaller migration ($m$) decreases the probability. Other parameters: $N = 10^4$ and $\alpha_0 = \alpha_1 = 0$.
(TIF)

## Acknowledgments

We thank the *Evolutionary Theory Department* in the MPI Plön for the methodological insight and the *CRC 1182: Origins and Functions of Metaorganisms*, in particular to Ute Hentschel, Hinrich Schulenburg, and Florence Bansept for fruitful discussions. Also to the anonymous reviewers for the feedback to improve the manuscript.

## Author Contributions

**Conceptualization:** Román Zapién-Campos, Michael Sieber, Arne Traulsen.

**Formal analysis:** Román Zapién-Campos, Michael Sieber, Arne Traulsen.

**Investigation:** Román Zapién-Campos.

**Software:** Román Zapién-Campos.

**Supervision:** Arne Traulsen.

**Writing – original draft:** Román Zapién-Campos, Michael Sieber, Arne Traulsen.

**Writing – review & editing:** Román Zapién-Campos, Michael Sieber, Arne Traulsen.

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
