## [Decision Letter · Decision Letter 0]

18 May 2020

Dear Dr. Traulsen,

Thank you very much for submitting your manuscript "Stochastic colonization of hosts with a finite lifespan can drive individual host microbes out of equilibrium" for consideration at PLOS Computational Biology.

As with all papers reviewed by the journal, your manuscript was reviewed by members of the editorial board and by several independent reviewers. In light of the reviews (below this email), we would like to invite the resubmission of a significantly-revised version that takes into account the reviewers' comments.

We cannot make any decision about publication until we have seen the revised manuscript and your response to the reviewers' comments. Your revised manuscript is also likely to be sent to reviewers for further evaluation.

Sincerely,

Dominik Wodarz

Associate Editor

PLOS Computational Biology

Natalia Komarova

Deputy Editor

PLOS Computational Biology

Reviewer's Responses to Questions

**Comments to the Authors:**

Reviewer #1: The review is uploaded as attachment.

Reviewer #2: In this manuscript, the authors present an expanded (near) neutral model for microbial community assembly in hosts of varying lifespans, to explain how patterns of microbial colonization and co-existence can emerge from ecological drift when the timescale of the host is considered. Using a combination of model analytics and simulations, the authors present a set of interesting insights into the microbial community assembly process within a host, particularly in clarifying the relationship between lifespan and migration in determining convergence to the infinite-lifespan case. This manuscript makes a convincing case for considering microbiomes as an assembly process rather than an equilibrium scenario, and the model presented will provide a useful starting point for considering further expansions into the near-neutral scenario where interactions among microbes are permitted to vary.

It would be interesting if the authors were to speculate further on the possible uses and expansions of this model. For example, in the Discussion, priority effects are mentioned but not expanded upon. The model makes predictions about what the distribution of colonization intensities/frequency of unoccupied individuals should look like for an ordinary Markov process on a population of identical hosts – expansions could be used to make predictions for departures from the current model in any of these assumptions. Some of the interesting cases might involve the question of memory (e.g. priority effects). For example, microbial “death” might be bursty (as seems to be the case in zebrafish) – can this still be a Markov process (maybe a Levy walk?), or is it necessary to allow the system to remember prior states? How can we tell these possibilities apart? These expansions are beyond the scope of the current work, but increasing the future work component of the Discussion along the lines of this or other biologically relevant scenarios would improve the accessibility of the manuscript.

Minor points

Define pi in text before presentation of equation 1.

4D-F - It would be informative to plot out (as a supplement) the heatmap of xi values that correspond to these plots.

**Have all data underlying the figures and results presented in the manuscript been provided?**

Reviewer #1: None

Reviewer #2: No: Include simulation code or github

PLOS authors have the option to publish the peer review history of their article (what does this mean?). If published, this will include your full peer review and any attached files.

Reviewer #1: No

Reviewer #2: Yes: Nic Vega
---

## [Decision Letter · Decision Letter 1]

10 Aug 2020

Dear Dr. Traulsen,

Thank you very much for submitting your manuscript "Stochastic colonization of hosts with a finite lifespan can drive individual host microbes out of equilibrium" for consideration at PLOS Computational Biology. As with all papers reviewed by the journal, your manuscript was reviewed by members of the editorial board and by several independent reviewers. The reviewers appreciated the attention to an important topic. Based on the reviews, we are likely to accept this manuscript for publication, providing that you modify the manuscript according to the review recommendations.

The reviewer thought that their comments were adequately taken into account, but had some remaining comments that they requested to have addressed.   

Sincerely,

Dominik Wodarz

Associate Editor

PLOS Computational Biology

Natalia Komarova

Deputy Editor

PLOS Computational Biology

[LINK]

Reviewer's Responses to Questions

**Comments to the Authors:**

Reviewer #1: The review is attached as a pdf file.

**Have all data underlying the figures and results presented in the manuscript been provided?**

Reviewer #1: None

PLOS authors have the option to publish the peer review history of their article (what does this mean?). If published, this will include your full peer review and any attached files.

Reviewer #1: No
---

## [Editor Report · Decision Letter 2]

2 Sep 2020

Dear Dr. Traulsen,

We are pleased to inform you that your manuscript 'Stochastic colonization of hosts with a finite lifespan can drive individual host microbes out of equilibrium' has been provisionally accepted for publication in PLOS Computational Biology.

Best regards,

Dominik Wodarz

Associate Editor

PLOS Computational Biology

Natalia Komarova

Deputy Editor

PLOS Computational Biology

---

## [Editor Report · Acceptance letter]

28 Oct 2020

PCOMPBIOL-D-20-00223R2 

Stochastic colonization of hosts with a finite lifespan can drive individual host microbes out of equilibrium

Dear Dr Traulsen,

I am pleased to inform you that your manuscript has been formally accepted for publication in PLOS Computational Biology. Your manuscript is now with our production department and you will be notified of the publication date in due course.

With kind regards,

Nicola Davies
